# Conversion of Sugar Di-Ketals to Bio-Hydrocarbons through Catalytic Cracking over Beta Catalysts in Fixed and Fluidized Catalytic Beds

Cristiane Cardoso [1,*], Yiu L. Lam [1], Marlon B. B. de Almeida [2] and Marcelo Maciel Pereira [1,*]

1   Instituto de Química, Universidade Federal do Rio de Janeiro (UFRJ), Av. Athos da Silveira Ramos, No. 149, Bloco A, Centro de Tecnologia, Cidade Universitária, Rio de Janeiro 21941-909, RJ, Brazil
2   PETROBRAS, Centro de Pesquisas e Desenvolvimento Leopoldo A. Miguez de Mello (CENPES), Av. Horácio Macedo, 950, Cidade Universitária, Ilha Do Fundão, Rio de Janeiro 21941-915, RJ, Brazil
*   Correspondence: cristianecardoso@iq.ufrj.br (C.C.); maciel@iq.ufrj.br (M.M.P.)

**Abstract:** Second-generation biomass (BM) can be produced in amounts that meet worldwide fuel demands. However, BM favors parallel and undesirable reactions in its transformation chain. We circumvent this problem by first modifying BM by ketalization, giving a user-friendly liquid we named BP (bio-petroleum). This study converted a representative compound of BP, DX (1,2:3,5-di-O-isopropylidene-$\alpha$-D-xylofuranose), mixed with n-hexane by beta zeolites and catalysts containing beta zeolite. Beta zeolite showed low coke and high liquid product yields in converting this mixture (having 30 wt. % DX) into hydrocarbons in a fixed-bed reactor at 500 °C with a space velocity of 16 h$^{-1}$ (0.3 catalyst/feed). Its performance was further improved by steam treatment (lowering the coke yield by lowering the acid site density) or incorporation into a catalyst (improving DX participation due to the active sites in the matrix). Further, by changing the conversion process from a fixed bed to a fluidized cracking unit, a much larger amount of the deactivated catalyst could be used (catalyst/feed = 3), remarkably reducing oxygenates and fully converting DX. Additionally, the green hydrocarbon efficiency (olefin, aromatics, furans, and cyclo-alkanes) of DX was approximately 77%. Hence, beta catalysts were shown to have a great potential to provide green fuels for future bio-refineries.

**Keywords:** bio-hydrocarbons; sugar acetals; biomass; beta zeolite; catalyst and steam treatment

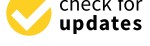



## 1. Introduction

Motivated by the challenge to use second-generation biomass as a renewable source of fuels and petrochemicals, we have introduced a method of transforming typical sources such as sugarcane bagasse, wood, and paper residues by using a mild ketalization process that yields a user-friendly liquid composed of acetyl derivatives of carbohydrates. This product is named bio-petroleum, BP [1,2], in contrast to bio-oils, the liquid products obtained from the pyrolysis of biomass [3]. In short, the ketalization process is carried out by processing second-generation biomass (sugar cane bagasse in our case) in the presence of large amounts of ketones in a temperature range of 100–180 °C and a small amount of acid as a catalyst.

Table 1 compares the general properties of BP with those of pyrolysis bio-oil [2,4–6]. Aside from being neutral and stable during storage, due to the protection of the active functional groups of the biomass, BP shows further advantages when subjected to further processing. Two key advantages are a low coke yield and the high incorporation of green carbon into the desired products.

**Table 1.** A typical bio-oil from thermal pyrolysis compared to bio-petroleum from ketalization.

| | Bio-Oil (Thermal Pyrolysis) [4–8] | BP (Ketalization Ex) [2] |
|---|---|---|
| General properties | Varied greatly as a function of source | |
| Acidity | High, pH 2 to 4 [4,5,7] | Almost neutral |
| Viscosity | High due to heavy oxygenates, examples varied: 25 to 1000 [4], 400 to 500 [6], 10 to 100 cp [7] (measured at 25 to 60 °C) | 200 cP |
| Stability (room T storage) | Unstable due to polymerization and condensation of components [4,5] | Stable after 4 months |
| Miscibility with solvents | Not miscible with most solvents | Infinitely miscible with acetone, partially with non-polar solvents and VGO |
| Composition (liq. Product) | | |
| CHO content | 55: 6: 38; typically, O 35 to 40% [5,6] | 48 to 55: 6.4 to 7: 46 to 37 [2] |
| Typical components (in wt. %) | Contains hundreds of compounds in 2 phases: aqueous phase, 15 to 30%, contains acetone, acetic acid MeOH; organic phase: 30 to 40% oxygenates, contains hydroxyl ketones/aldehydes, phenols, furans, acids, and sugars, 10 to 20% hydrocarbons [4–6,8] | 50% monoketals 30% higher ketals 20% lignin derivatives |

These advantages were demonstrated by some recent publications, which employed an important component of bio-petroleum, 1,2-3,5-O-di-isopropylidene-D-xylofuranose (DX), as a model for tests. N-hexane was chosen as a co-reagent, since it mixes well with DX; being a short saturated hydrocarbon, it provides much less competition for DX to access the active sites. Further, it is much less reactive and gives fewer products in the catalytic cracking reaction. DX (mixture with n-hexane) was converted in fixed and fluidized beds using ZSM-5, USY, and typical FCC catalysts, and in all cases, low coke and high yields in the gasoline fraction rich in aromatics were obtained [9–11]. In hydro-deoxygenation reactions under mild hydrogen pressure, using Pd and Pd-Cu catalysts supported on ZSM-5 and beta zeolites, these DX/n-hexane mixtures gave significant fractions of hydrocarbons of higher molecular weight (C10+), along with a low coke yield [12,13]. It is noteworthy to point out that the advantage of using DX as a model test could simplify the product analysis, give clearer insights into the reaction steps in the ketal transformation, and hence suggest a more definite correlation between catalyst properties and product yields. Using either DX or BP, with compositions described in Table S1, as reactants, very similar catalytic results were observed in a variety of tests. These results are the subjects of future publications [14,15].

In the cracking of DX/n-hexane mixtures in a fixed bed, beta zeolites showed better performance than ZSM-5 and USY zeolites under the same conditions [16]: a higher liquid product yield and a compromise between the gaseous product and coke yield. As the beta zeolite employed has intermediate properties between ZSM-5 and USY, these results may be interpreted as a compromise between the pore structure and acidity. In this work, we wanted to explore further beta and related catalysts to crack this sugar acetal. The objective is to process a greater amount of this biomass derivative in the co-feed, with a maximum liquid hydrocarbon yield, high deoxygenation and the maintenance of a low coke yield in the process. The catalyst properties being evaluated were the acid site density, the presence of a meso-area matrix and the catalyst-to-feed ratio.

Aside from this practical aim, from the product analysis under different conversion conditions, we also speculate on the sequential steps involved in the formation of hydrocarbon products from the sugar acetal and, hence, the role of the catalyst in these steps.

## 2. Materials and Methods

Preparation of Catalysts. Beta zeolite ($SiO_2/Al_2O_3$ mole ratio 25, cod CP-814E) was purchased from Zeolyst International Inc. (Pennsylvania, USA). It was calcinated at 550 °C for 3 h and named HBEA. A catalyst containing the same beta zeolite (CP-814E) and a

matrix of silica and alumina was provided by Petrobras and named the beta catalyst. It was calcined under the same conditions and named AD. To obtain samples with fewer acid sites, hydrothermal treatment was carried out in a muffle furnace equipped with a deionized water vapor saturator (100%) at a volumetric flow rate of 1 mL min$^{-1}$. Beta zeolite and the beta catalyst were heated from room temperature with a heating ramp of 10 °C min$^{-1}$ up to 720 °C and were kept at this final temperature for 2 h, giving samples named DHBEA and DAD, respectively. They are referred to as deactivated samples.

Characterization of the catalysts. The crystallinity of beta zeolites in all catalysts was determined by X-ray diffraction (XRD) performed using a diffractometer from Rigaku (Japan) Ultima IV, (Cu Kα λ = 0.1542 nm) a fixed energy source (40 kV and 20 mA) and a scanning rate of 0.02° s$^{-1}$ at intervals from 2θ from 5° to 80°. The crystallographic records were obtained using the Rigaku PDXL program, version 2.3.1.0.

The textural properties were measured by nitrogen physisorption performed at −196 °C in an ASAP 2420 device from Micromeritics (Georgia, USA). Before the analysis, the samples were pre-treated at 300 °C under vacuum for 15 h and then subjected to degassing at 150 °C for 1 h. The specific surface area (SBET) and external specific area (Sext) was calculated through Brunauer–Emmett–Teller (BET) and t-plot methods, respectively. The total pore volume was determined at 0.98P/P$_o$, and the micropore volume was determined from the t-plot.

Solid-state magic-angle-spinning NMR (MAS NMR) measurements were performed on a 400 Ultra Shield from Bruker (Germany). The samples were dried in an oven at 105 °C overnight. For $^{27}$Al MAS NMR, the samples were packed into a 2.5 mm ZrO$_2$ rotor. The spectra were recorded at a resonance frequency of 104.29 MHz. The rotor was spun at 20 kHz, and 4000 average scans were obtained for each spectrum. The spectra were normalized to the sample mass for quantitative comparison. For $^{29}$Si MAS NMR, the samples were packed into a 2.5 mm ZrO$_2$ rotor. The spectra were recorded at a resonance frequency of 99.3 MHz. The rotor was spun at 20 kHz with 15,000 average scans for each spectrum. The spectra were normalized to the sample mass for quantitative comparison.

Synthesis of DX. 1,2:3,5-Di-O-isopropylidene-α-D-xylofuranose (DX) was synthesized via the reaction of D-xylose (98.5%, VETEC,30 g) with acetone (99%, VETEC, 800 mL). The reaction mixture was magnetically stirred, cooled in an ice bath, and held at 10 °C. Then, 20 mL of H$_2$SO$_4$ (P.A., VETEC) was added dropwise to the cooled suspension for 15 min, and then the system was heated to 20 °C under magnetic stirring for 90 min. The mixture was cooled to 10 °C and neutralized through the dropwise addition of 80 mL of a NaOH (40 wt. %) aqueous solution. The resultant suspension was filtered under vacuum, and the filtrate was introduced into a rotary evaporator to remove acetone under low pressure at 55 °C. From this step, a white emulsion residue was obtained that was mixed with 120 mL of ethyl acetate, forming two phases. The aqueous phase was separated, and the organic phase was washed two times with 15 mL of H$_2$O and introduced into a rotary evaporator in order to evaporate ethyl acetate under low pressure at 35 °C. The residual transparent oil (consisting of DX and residual impurities) was then washed with n-hexane in order to extract DX. Afterward, n-hexane was evaporated under reduced pressure, and DX was isolated. Then, it was dissolved once more in n-hexane to obtain the mixture used in the reactions, 30 wt. % DX in n-hexane. This mixture was kept at 5 °C to avoid n-hexane evaporation and DX degradation. The purity of DX was verified through paper chromatography and GCMS analyses.

Catalytic Cracking. For tests carried out in a fixed-bed catalytic cracking unit (FB) (as presented in Figure S1), 3 mL of the reactant (pure n-hexane or 30 wt. % DX in n-hexane) was injected for 15 min over 500 mg of the catalyst, with nitrogen as a carrier gas at a flow of 100 mL min$^{-1}$ (calibrated at room temperature). The reactor was operated at 500 °C.

For tests carried out in a laboratory fluidized-bed catalytic cracking unit (FCC), as presented in (Figure S2), the catalyst was first activated in a nitrogen atmosphere for 12 h at 500 °C. Then, 3 mL of the reactant (pure n-hexane or 30 wt. % DX in n-hexane) was injected for 0.5 min over 15 g of the catalyst. A 200 mL min$^{-1}$ nitrogen flow was used to

ensure catalyst fluidization and calibrated at the reaction temperature. The reactor was also operated at 500 °C.

The reaction products were distributed in gas, liquid, and coke; DX also produces water, as described below. The organic liquid products (LPs) were hydrocarbons and oxygenates.

The volume of gas produced during the reaction was determined by the displacement of pure water in an Erlenmeyer of a saturated aqueous NaCl solution (by the difference in the water mass before and after the reaction). For the fixed-bed tests, the gas composition ($H_2$, CO, $CO_2$, methane, and hydrocarbon up to $C_4$) was analyzed online every 2.5 min of the reaction, using the average of 4 samples injected into a Micro GC 490 gas chromatograph with a TCD detector from Agilent Technologies (California, USA). The gas composition varied less than 20% during the time on stream. For the FCC tests, the average composition was determined.

The organic liquid fraction (LP) was obtained through condensation (−10 °C) by means of a condenser placed right after the reactor exit. The liquid amount was obtained by the weight difference of the condenser before and after the reaction. The liquid fraction was analyzed by GCMS to identify the products and GCFID to quantify them, and the equipment used was from Agilent Technologies (California, USA). The CG system is an 7890A CG coupled to both a 5975C MS in electron impact mode and an FID detector, and an Agilent HP-5MS column was used; the oven temperature was maintained at 30 °C for 7 min, followed by a ramp to 170 °C for 40 min, and helium was used as the carrier gas. At the inlet, a 20:1 split ratio, 14 psi and 290 °C were used. All samples were injected without dilution.

The calculations will be illustrated below to demonstrate the effects obtained in the experiments.

The liquid hydrocarbon products were obtained by subtracting the mass of unreacted n-hexane from the total liquid mass. The determination of unreacted n-hexane in the liquid product is given in the Supplementary Materials.

The mass fraction of each type of product of interest was obtained by multiplying the FID area (%) by the respective chromatographic factor, given in Table S2 in the Supplementary Materials.

Water is one of the products of the reaction but could not be readily determined due to absorption by the zeolite carried by the vapor and loss on the walls of the reactor. Hence, we determined the water produced on the basis of the oxygen balance, given by Equation (1) below:

$$\text{Oxygen from product water } = \text{ODX} - \text{OL} - \text{OG} \tag{1}$$

ODX, OL, and OG are oxygen from DX, from the liquid phase, and from the gas phase. If the oxygenate amount in the liquid product was 2% or less, OL was simply neglected. The weight fractions of oxygen in CO, $CO_2$, the oxygenated liquid product, DX, and $H_2O$ were considered 0.53, 0.73, 0.15, 0.35, and 0.88, respectively.

Finally, the conversion of each reactant and the yield of product X are defined by Equations (2) and (3) below.

$$\text{Conversion (\%)} = \frac{\text{initial wt. of reactant } - \text{ final wt. of reactant}}{\text{initial wt. of reactant}} \times 100 \tag{2}$$

$$\text{Yield X (\%)} = \frac{\text{wt. of product X}}{\text{initial wt. of reactant}} \times 100 \tag{3}$$

The amount of coke in the used catalysts was determined using a Netzsch TG-IRIS thermogravimetry device. The samples were heated from 35 °C to 250 °C at a rate of 10 °C min$^{-1}$ under a flow of $N_2$ (30 mL min$^{-1}$). The temperature was maintained at 250 °C for 30 min, after which the flow was changed to synthetic air (20.9% $O_2$ in $N_2$), and the temperature was increased to 700 °C at a rate of 10 °C min$^{-1}$ and then maintained at

700 °C for 30 min. The amount of coke in the catalyst corresponded to the weight loss at temperatures above 250 °C, and the coke yield was estimated by Equation (3).

## 3. Results and Discussion

### 3.1. Catalyst Properties

The physicochemical properties of beta zeolite and the beta catalyst, both fresh and deactivated, are presented in Table 2. The position and intensity of the reflections of diffractograms of beta zeolite and the catalyst were compared with data provided by JCPDS (Joint Committee on Powder Diffraction Standards) reference cards from the International Center for Diffraction Data (ICDD) library. These showed typical diffractograms of beta zeolite (Figure 1a). The sum of the integrated intensities of the dominant peaks between 19.8° and 24.3° 2θ was used as an index of the sample, and the hydrothermally treated samples had a relative drop in crystallinity of approximately 33% (Table 2). This fact indicates that desilication with water vapor leads to the partial amorphization of beta, both as a zeolite and as a catalyst. However, the structures of the catalysts were maintained. Gusev et al. [17], who observed a reduction in crystallinity, reported that high resistance to steam treatment (770 °C—7 h) was observed for both pure ZSM-5 zeolite (ca. 12%) and the P/ZSM-5 catalyst (ca. 10%) under severe vaporization conditions.

**Table 2.** Structural and textural properties of beta zeolite and beta catalyst.

| Samples | $A_{BET}$ [a] $(m^2g^{-1})$ | $A_{ext}$ [b] $(m^2g^{-1})$ | $V_{micro}$ [b] $(cm^3g^{-1})$ | $V_{total}$ [c] $(cm^3g^{-1})$ | Relative Cryst. XRD (%) [d] | Total Acid Sites (μmol/g) [e] | SAR [f] (RMN) | $SiO_2$ [g] (%wt) | $Al_2O_3$ [g] (%wt) |
|---------|---------|---------|---------|---------|---------|---------|---------|---------|---------|
| AD | 250 | 193 | 0.039 | 0.38 | 100 | - | - | 45.3 | 51.9 |
| DAD | 161 | 132 | 0.013 | 0.35 | 66 | - | - | 43.6 | 54.3 |
| HBEA | 609 | 182 | 0.18 | 0.36 | 100 | 459 | 28 | 92.9 | 6.9 |
| DHBEA | 391 | 135 | 0.10 | 0.40 | 67 | 110 | 43 | 92.7 | 7.1 |

[a] BET area (BET method); [b] external surface area and micropore volume (T-plot method); [c] Vtotal—total volume determined from the volume absorbed at P/P0 = 0.9; [d] relative crystallinity for DHBEA and DAD calculations were based on fresh HBEA and AD samples treated as 100% crystalline, respectively.; [e] total acidic sites determined by Pyridine FTIR; [f] SiO2/Al2O3 ratio calculated by $^{29}$Si MAS NMR; [g] X-ray fluorescence data.

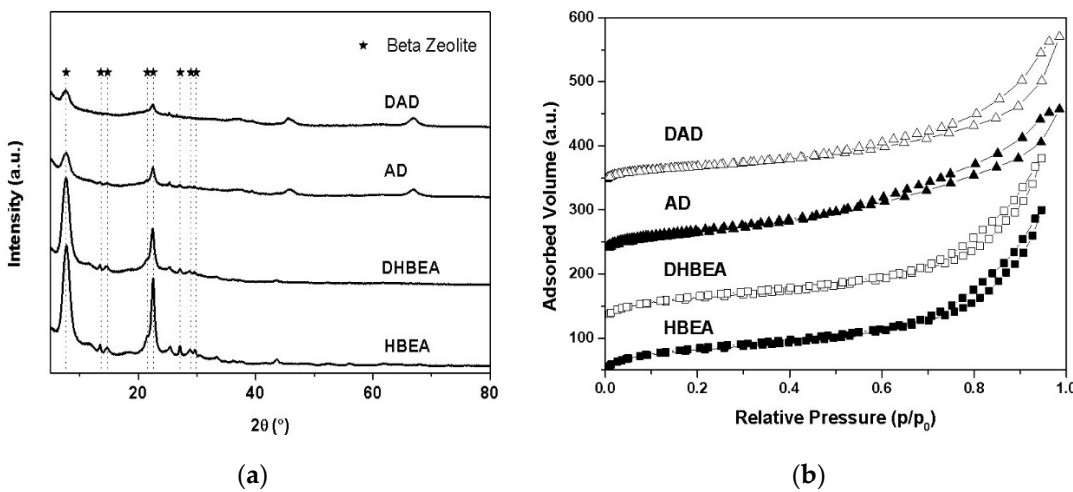

**Figure 1.** Characterization of beta zeolite and beta catalyst: (**a**) X-ray diffraction and (**b**) physisorption N₂ adsorption/desorption isotherm (various curves shifted upward for clarity). Legend: HBEA (Fresh beta zeolite); DHBEA (Hydrothermally treated beta zeolite); AD (Fresh beta catalyst) and DAD (Hydrothermally treated beta catalyst).

For the textural properties, one could note that even though the catalyst has a lower amount of beta zeolite (shown by the crystallinity and micropore volume measurements),

it shows a comparable external surface area and total pore volume. This indicates that the matrix components provide a high and accessible surface area. This feature will be invoked to explain some of the results of catalytic tests presented in later sections.

Upon steam treatment, both beta zeolite and the beta catalyst showed a reduction in the BET area, possibly due to the drop in the crystallinity of the zeolite and to the partial blockage of the pores by the aluminum extracted from the zeolite framework. As for the volume of micropores, the reduction after steam treatment relative to the fresh sample was smaller for the catalyst when compared to pure beta. It can be inferred that the physical–chemical properties of the catalyst matrix contributed to the retention of the micropore structure. The effect of hydrothermal treatment was less in the sample with a matrix present than in pure beta zeolite. This complex interaction has often been cited in the literature [18–21]. The isotherms are shown in Figure 1b.

$^{27}$Al MAS NMR measurements were performed on the zeolites. The increased silica/alumina ratio of the framework (SAR) confirms the reduction in the acidity of the zeolite by steam treatment (Table 2), similar to that observed by Maier et al. [22].

*3.2. Pure Zeolite in Fixed Bed: Reducing Acid Sites by Steam Treatment*

The catalytic performance of fresh and deactivated beta carried out in a fixed-bed unit is summarized in Table 2. The material balances were obtained by adding the liquid, gas, and coke fractions and reached around 95%. The results were normalized to 100% and reported in the tables to facilitate comparisons.

In the conversion of the DX/n-hexane mixture, DX reached almost 100% conversion to DHBEA even when the number of acid sites in beta decreased due to hydrothermal treatment. However, n-hexane conversion showed a decrease from fresh zeolite (HBEA) to deactivated zeolite (DHBEA). The decrease in the participation of n-hexane in the products and the increase in the ratio between the yield of the liquid product and the yield of the gaseous product are clearly shown in Table 2. These results are in line with the reduction in the number of active sites, as observed previously for other catalysts [10].

Hydrothermal treatment resulted in a 50% decrease in the yield of the gaseous fraction, especially reducing CO formation (Table S3 in the Supplementary Materials). The coke yield had a slight reduction. Hence, an increase in the liquid fraction resulted (Table 3).

**Table 3.** Conversions and yields of liquid, gas, coke, unreacted n-hexane, and water were obtained by catalytic cracking with a percentage of 30% DX/n-hexane in HBEA and DHBEA at 720 °C.

| | HBEA [a] | DHBEA [a] |
|---|---|---|
| Conversion (%) | | |
| n-Hexane conversion | 5.7 | 2.2 |
| DX conversion | 99.6 | 99.8 |
| Yield (wt. %) | | |
| Gas | 12.8 | 6.0 |
| Liquid | 84.3 | 91.2 |
| HC product [b] | 11.2 | 9.8 |
| Oxygenates | 2.0 | 5.4 |
| n-Hexane | 66.0 | 68.5 |
| Water [c] | 5.1 | 7.5 |
| Coke | 3.1 | 2.9 |

[a] When using 500 mg of zeolite and reaction temperature of 500 °C; [b] liquid hydrocarbon products, including paraffins, olefins, monoaromatics, polyaromatics, and unidentified products. [c] Oxygen balance data.

In the liquid fraction, the monoaromatic yield decreases with the hydrothermal treatment, while the oxygenates increase (Figure 2). The oxygenates from the 30 wt. % DX mixture were mainly identified as ketones and furans (Table S4 in the Supplementary Materials). Despite the high yield of oxygenates in the liquid product, water is the most important deoxygenation product. Thus, DX is mostly deoxygenated to produce water, without much loss of useful carbon.

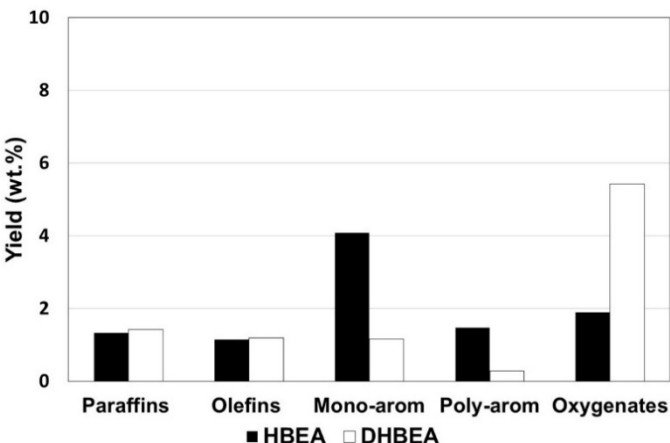

**Figure 2.** Yields of organic liquid products obtained by catalytic cracking of 30% DX/n-hexane from HBEA and DHBEA, 500 mg of zeolite at 500 °C. Legend: Mono-arom—monoaromatic; Poly-arom—polyaromatic; HBEA (Fresh beta zeolite); DHBEA (Hydrothermally treated beta zeolite).

The high conversion of DX indicates that the initial decomposition of DX does not require many acid sites, only those that remained after steam treatment. This decomposition resulted mainly in water formation. However, the further transformation of oxygenated intermediates may require such sites, indicated by the high oxygenate and low CO formation from the deactivated beta zeolites. Furthermore, the low yield in the monoaromatics can be interpreted mainly due to a decrease in the density of the acid sites [23].

However, the acidity effect of the overall transformation is complex. It is very likely that changes in the acid sites on catalysts alter their interaction not only with the reactants but also with the intermediate products formed. In addition, steam treatment can also change the amount and type of non-framework alumina. This, in turn, can limit or block reagent access and product desorption from active sites.

### 3.3. Catalysts in Fixed Bed: Comparison with Steam-Treated Beta Zeolite with Fresh and Steam-Treated Catalysts

The reduction in acidity in the cracking process achieved a reduction in gas and coke yields, but the deoxygenation is not satisfactory. Hence, we explored the use of formatted beta catalysts considering two possible contributions of the catalysts. First, the typical catalyst also contains matrix elements that may help in the initial transformation of large molecules and improve gasoline octane due to a lower hydrogen transfer rate vs. the cracking rate. However, active matrices also enhance low-selectivity cracking, leading to an increase in coke and dry gas, often at the expense of gasoline [24]. Next, catalysts could be used in the fluidized-bed process. This process allowed us to use a much higher amount of deactivated zeolite, increasing the total number of acid sites without increasing the acid site density. This section tested the catalysts in a fixed bed as a bridge to the test in a fluidized-bed unit, to be presented in the next section.

As observed in Table 4, the conversion of DX with the beta catalyst was 100% from both fresh and deactivated catalysts. Yet, deactivated beta zeolite did not completely convert DX. This fact may be associated with various factors. The active centers that make up the matrix of the catalyst may have collaborated to initiate the DX reaction. Additionally, since the catalyst has a mesopore volume 2 times greater than that of beta zeolite [25,26] (Table 2), its active sites may be more accessible.

**Table 4.** Performance comparison of hydrothermally deactivated beta zeolite and catalysts with 30%DX/n-hexane mixture at 500 °C, fixed bed.

|  | DHBEA | AD | DAD |
|---|---|---|---|
| Conversion (%) | | | |
| n-Hexane conversion | 2.2 | 0.0 | 0.0 |
| DX conversion | 99.8 | 100 | 100 |
| Yield (wt. %) | | | |
| Gas | 6.0 | 3.3 | 4.0 |
| Liquid | 91.2 | 92.8 | 93.3 |
| HC product [a] | 9.8 | 12.4 | 11.8 |
| Oxygenates | 5.4 | 1.1 | 3.6 |
| n-Hexane | 68.5 | 70.1 | 70.3 |
| Water [b] | 7.5 | 9.2 | 7.6 |
| Coke | 2.9 | 4.0 | 2.6 |

[a] Liquid hydrocarbon products, including paraffins, olefins, monoaromatics, polyaromatics, and unidentified products; [b] oxygen balance data.

The beta catalyst showed no detectable conversion of n-hexane, while even deactivated beta zeolite resulted in 2.2% conversion. This may be due to the fact that the beta catalyst contained less beta zeolite (~30%). Furthermore, DX and its intermediates may compete more successfully for active sites in the catalysts.

In the yield of the liquid, the reduction in the oxygenated product with the fresh catalyst (1.1% wt.) stands out compared to all samples tested so far. However, this greater deoxygenation of DX with the catalyst caused a slight increase in coke compared to the DHBEA results (Table 4). Aromatics were the majority in the liquid product of the fresh catalyst (Figure 3) and were classified mainly as naphthalenes and aromatics with eight carbons.

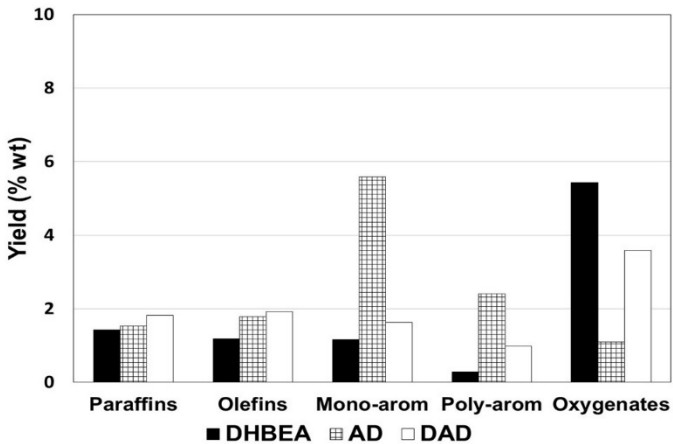

**Figure 3.** Yields of organic liquid products obtained by cracking 30% DX/n-hexane over DHBEA, AD, and DAD, 500 mg of catalyst at 500 °C. Legend: Mono-arom—monoaromatics; Poly-arom—polyaromatic; DHBEA (Hydrothermally treated beta zeolite); AD (Fresh beta catalyst) and DAD (Hydrothermally treated beta catalyst).

As there was no conversion of n-hexane with the fresh and steam-treated catalysts, the gaseous product was formed from the conversion of DX. Here, we observed that DX did contribute to the formation of methane but produced negligible amounts of $C_3$ and $C_4$. A major part of the gaseous product is CO, with a much smaller amount of $CO_2$ (Table 5). Thus, DX and especially its intermediate oxygenated products underwent decarbonylation [23] and decarboxylation [24] reactions that eventually gave hydrocarbons.

**Table 5.** Yields of gases by percentage of DX obtained by catalytic cracking of 30% DX in n-hexane on beta zeolite, beta catalyst hydrothermally treated at 720 °C, and fresh beta catalyst, 500 mg of catalyst at 500 °C of reaction.

| | Gas (%wt) | | | | | | | | | | |
|---|---|---|---|---|---|---|---|---|---|---|---|
| | $H_2$ | $CH_4$ | $C_2H_4$ | $C_2H_6$ | $C_3H_8$ | $C_3H_6$ | $C_4H_{10}$ | $C_4H_8$ | CO | $CO_2$ | Total |
| DHBEA | 0.1 | 2.1 | 0.2 | 0.0 | 0.0 | 0.1 | 0.0 | 0.0 | 2.9 | 0.6 | 6.0 |
| AD | 0.1 | 1.4 | 0.0 | 0.0 | 0.0 | 0.2 | 0.0 | 0.0 | 1.5 | 0.0 | 3.3 |
| DAD | 0.0 | 1.2 | 0.0 | 0.0 | 0.0 | 0.1 | 0.0 | 0.0 | 2.6 | 0.1 | 4.0 |

As observed from the steam treatment of pure beta zeolite in the previous section, the coke yield of the deactivated catalyst was less than that of the fresh catalyst, although the decrease in active sites may have limited the cracking of intermediate products of DX. Hence, to increase the transformation of the intermediate products of the reaction, a greater number of sites is still necessary for the complete deoxygenation of DX. An alternative would be the use of an FCC unit, which will be presented next.

*3.4. Hydrothermally Treated Catalyst in Fluidized Bed vs. in Fixed Bed*

In this analysis, a brief comparison of the steam-treated beta catalyst (DAD) was made between the results obtained with experiments in the fixed-bed unit (FB) and those originating from the fluidized-bed unit (FCC) [27]. A variation in the catalyst/load ratio (cat/oil) from 0.3 in the FB unit to 3 in the FCC unit was used, a 10-fold increase. Table 6 summarizes the results obtained with 30% DX in n-hexane in both units. Figure 4 compares the organic liquid product distribution obtained from these units.

Under the reaction conditions adopted in FCC, with the high amount of catalyst, n-hexane still had great interference in the results of the conversion of the DX/n-hexane mixture. One can note in Table 4 that n-hexane showed 7% conversion and higher amounts of gas products. The greater amount of liquid hydrocarbon product (13.6% $w/w$) may also include isomerized n-hexane. However, the catalytic system in FCC did result in much more significant deoxygenation. A lower yield of oxygenates in the liquid (0.9% $w/w$) is observed in Table 6. The main oxygenated components were furans, ketones, and phenols (Table S5 in the Supplementary Materials). Additionally, greater amounts of oxygenated gaseous products (CO, $CO_2$) were detected (Table S6 in the Supplementary Materials). Similar amounts of water were formed. Hence, FCC enabled the better transformation of DX to hydrocarbons with an increasing amount of the low-activity catalyst.

**Table 6.** Hydrothermally treated beta catalyst (DAD) with different cat/oil ratios for fixed-bed and fluidized-bed experiments.

| Cracking Unit | FB [a] | FCC [b] |
|---|---|---|
| Ratio cat/oil | 0.3 | 3 |
| Conversion (%) | | |
| n-Hexane conversion | 0.0 | 7.1 |
| DX conversion | 100 | 100 |
| Yield (wt. %) | | |
| Gas | 4.0 | 8.0 |
| Liquid | 93.3 | 87.1 |
| Liquid product [c] | 11.8 | 13.6 |
| Oxygenated | 3.6 | 0.9 |
| n-Hexane | 70.3 | 65.1 |
| Water [d] | 7.6 | 7.5 |
| Coke | 2.6 | 4.9 |

[a] Run time in fixed-bed unit (FB): 15 min; [b] run time in fluidized-bed unit (FCC): 0.5 min; [c] liquid hydrocarbon products, including paraffins, olefins, monoaromatics, polyaromatics, and unidentified products; [d] oxygen balance data.

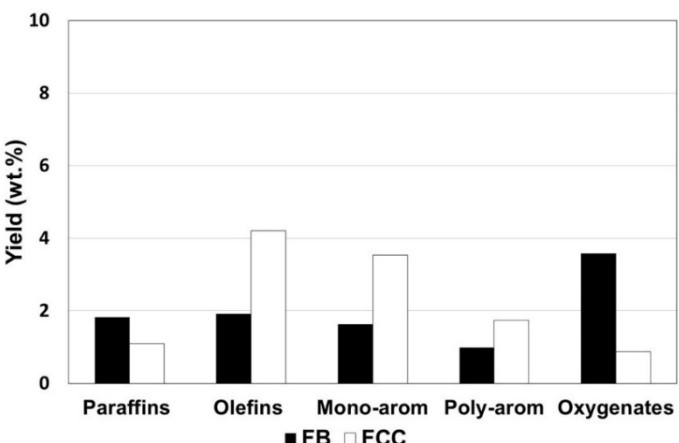

**Figure 4.** Yields of organic liquid products obtained by cracking 30% DX/n-hexane in hydrothermally treated beta catalyst at 500 °C. Legend: Mono-arom—monoaromatic; Poly-arom—polyaromatic; FB (Fixed Bed) and FCC (Fluid Catalytic Cracking).

*3.5. Outlook*

Firstly, we will further comment on the differences between the two testing processes. Both FB and FCC processes demonstrated successful DX conversion into green hydrocarbons. They also showed the benefits of applying a thermal deactivation treatment to the beta catalyst, used in its pure form or as the main active component in a catalyst. Both reaction modes can be used complementarily to develop and design catalysts to improve DX and, further, BP conversion into target products. FB is less time-consuming compared to FCC. Thus, FB can be used as an initial screening for catalyst properties. FCC usually reduces external diffusion and pressure build-up, but it is more time-consuming than FB [8]. In the cases compared, better deoxygenation, without much of a coke penalty, was obtained in FCC. Despite the high deoxygenation, these compounds are still present in the conditions used (30 wt. % DX and catalyst/feed = 3). However, the catalyst/feed ratio can be adjusted to values as high as 7–8 in FCC. In a separate study, we further varied certain reaction conditions in the FCC process, such as the catalyst/DX ratio, the injection time, and also % DX in the feed mixture [13]. This later study focusing on the process parameters, not on the catalyst properties, will be reported in another publication.

On the other hand, it is expected that, under certain conditions, FB can be used to convert BP. For example, when one uses catalysts with less activity, the FB unit may operate with a higher catalyst/load ratio if one could solve the problems with diffusion and pressure build-up. We are testing this possibility by using catalysts with different physical forms, such as pellets or extrudates. In short, we aim to use BP to produce drop-in fractions of green hydrocarbons in both the FB and FCC conversion modes.

Secondly, we will point out that DX conversion shows several important advantages. DX gives coke yields limited to 10 wt. % of DX. This value can be reduced by exploring the catalyst properties. DX gives low amounts of oxygenates and can in fact produce a liquid fraction that is fully deoxygenated. Additionally, DX gives a remarkably lower amount of phenols (30 wt. % DX converted in FCC, yield 0.01 wt. % phenol and 0.01 wt. % phenol derivatives). Further, DX mainly produced furan and ketone derivatives (Tables S4 and S5 in the Supplementary Materials). These compounds can be used to improve the gasoline octane number [27] or as chemical platforms [28]. Herein, we converted DX at a moderate concentration, but it was demonstrated previously that it could be converted at concentrations as high as 70 wt. % in FCC.

Now, let us discuss the overall picture linking our previous results of renewable production from biomass, extending the results of DX to bio-petroleum, BP. The biomass transformation chain into hydrocarbons occurs in two steps: 1—from biomass to BP with no carbon loss; 2—from BP, a stable, non-acidic chemical platform, to hydrocarbons.

Initially, DX was used as a feed or a representative model compound of BP, as it is present in BP in amounts up to 30 wt. % [2]. Our further experimental results comparing the conversion of DX and BP under comparable conditions indeed showed that the BP transformation closely parallels that of DX, not only in the cracking process but also in hydro-deoxygenation processes [11,13,16]. From one of these studies, work on co-processing BP with VGO over equilibrium catalysts, with mixtures of up to 75% BP, is being submitted for publication.

Though the biomass-conversion-to-fuel process should not be compared only for the conversion of derivatives, it is worth mentioning that conversions of bio-oil in typical FCC conditions generally leave high amounts of acidic and phenolic oxygenates. Further, bio-oil co-processed with mineral oil is usually converted in concentrations up to 20 wt. %. [28–30].

## 4. Conclusions

It was demonstrated that beta zeolite can produce bio-hydrocarbons with high added value from DX in a fixed bed and fluidized bed, a model di-acetal of bio-petroleum from sugarcane bagasse. Three properties of the catalytic system containing beta zeolites were examined.

Reducing the acidity of beta by applying hydrothermal treatment led to a decrease in interference from n-hexane in the mixture with DX and less gas and coke generation. However, DX cannot be totally converted into hydrocarbons in the fixed-bed unit by the deactivated zeolite.

The beta catalyst performed better than pure zeolite in the fixed-bed unit, with even less interference from the co-feed n-hexane and a further decrease in the yield of oxygenates. However, the coke yield slightly increased. Both of these observations could be attributed to the matrix participation in the DX transformation.

Deoxygenation had a significant improvement when a large amount of the deactivated catalyst was used in an FCC unit. Hence, these results indicate the great potential of the beta-containing catalyst in the conversion of bulky molecules, such as those observed in BP, for use in a bio-refinery.

## 5. Patents

Innovation Privilege. Registration number: BR1020190075880, title: "'PROCESS FOR CONVERTING SUGAR ACETALS INTO GREEN AROMATICS FOR APPLICATION AS PETROCHEMICAL AND FUEL INPUTS'"; registration institution: INPI—National Institute of Industrial Property. Deposit: 11/13/2018.

**Supplementary Materials:** The following supporting information can be downloaded at https://www.mdpi.com/article/10.3390/chemistry5010035/s1: Figure S1. Schematic of the fixed-bed unit used in the DX/n-hexane experiments; Figure S2. Schematic of the fluid catalytic cracking (FCC) unit used in the DX/n-hexane experiments; Table S1. Table of identified compounds of the general composition of bio-petroleum (BP) from LC-HRMS analysis in positive ESI mode; normalized intensity (a.u.) area; Table S2. Chromatographic factors employed for % mass of product of interest; Table S3. Gas yields obtained by catalytic cracking with 30% percentage change in DX/n-hexane in HBEA nd DHBEA; Table S4. Possibility of oxygenated compounds produced from DX/n-hexane reactions in the catalytic cracking of beta zeolite with steam treatment at 720 °C (DHBEA); Table S5. Possibility of oxygenated compounds produced from DX/n-hexane reactions in the catalytic cracking of the beta catalyst with steam treatment at 720 °C (DAD); Table S6. Yields of gas by percentage of DX obtained by catalytic cracking of n-hexane in FB and FCC, 30%DX/n-hexane in hydrothermal treatment at 720 °C on beta catalyst, 500 mg of zeolite at 500 °C.

**Author Contributions:** C.C. carried out all experiments, prepared the figures, and wrote the manuscript; M.B.B.d.A. prepared and characterized the catalyst; Y.L.L. and M.M.P. conducted the research. All authors were involved in the preparation of the final version of the manuscript. All authors have read and agreed to the published version of the manuscript.

**Funding:** This work was supported by CAPES (Coordenação de aperfeiçoamento de pessoal de nível superior), process 88881.189032/2018-01, which granted a sandwich scholarship, and FAPERJ (Fundação Carlos Chagas Filho de Amparo à Pesquisa do Estado do Rio de Janeiro) (contracts 210.068/2020 and E-26/210.799/2021).

**Data Availability Statement:** Not applicable.

**Conflicts of Interest:** The authors declare no conflict of interest.

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
