# Peer review of "Conversion of Sugar Di-Ketals to Bio-Hydrocarbons through Catalytic Cracking over Beta Catalysts in Fixed and Fluidized Catalytic Beds"

_chemistry, doi:10.3390/chemistry5010035_

Round 1

Reviewer 1 Report

This manuscript deals with obtaining hydrocarbons by catalytic cracking of biomass, previously subjected to a ketalization process using beta zeolites and some additives containing beta zeolite as catalysts. Given that the experimental work is correctly carried out, well discussed with an up-to-date bibliography, and the subject addressed is absolutely topical, this manuscript can be accepted for publication.

Author Response

We would like to thank and are very happy to have positive and constructive opinions from the reviewer

Reviewer 2 Report

The topics is important and the contribution is original. While the scientific level is about average the research presented by the manuscript is well connected to the application. So, I recommend it for publication in the present form.

Author Response

We would like to thank and are very happy to have positive and constructive opinions from the reviewer.

Reviewer 3 Report

The manuscript studies the catalytic cracking of a model compound for the ketalization of biomass over a beta zeolite catalyst. The subject is very interesting and the results can be potentially interesting. However, the manuscript requires major revision for publication. The issues to improve are the following:

- Presumably, the process of biomass ketalization is unknown process for the readers of the journal and should therefore be briefly explained.

- In addition, the composition of the bio-petroleum (strange notation) should be explained in greater detail and the selection of DX as a model justified.

- The purpose of using n-hexane together with DX should also be explained.

- Likewise, the objective of including the so-called “additive” should be explained, because it seems to be the catalyst embedded in a mesoporous matrix, which is the configuration of FCC catalysts.

- The purpose of steaming the FCC catalyst is to condition its porous structure and acidity, equilibrating the catalyst so that it reproduces its performance (with high gasoline yield) in reaction-regeneration cycles. Consequently, the treated catalyst is called "equilibrated" or "stabilized". “Deactivated” is the universal denomination of the spent catalysts after their use.

- The performance of catalysts with different zeolite has been studied in cracking bio-oil (Ibarra et al., J. Ind. Eng. Chem., 78 (2019) 372-382). This is a good reference on the effect of shape selectivity and acidity on product distribution.

- If the authors have information on the acidity of the catalysts, they should include it, because it is essential for understanding the results.

- The characteristics of the fluidized reactor should be better explained. And the reasons for the different results in fixed- and fluidized-bed reactors should also be explained.

- The role of water is very important to attenuate the deactivation of the catalyst in bio-oil cracking (Ibarra et al., Energy Fuels 33 (2019) 7458-7465). Consequently, a high deposition of coke can be expected. Do the authors have information on this regard?

- The experiments have been carried out with catalyst excess. Consequently, the conversion is almost 100% and the deactivation cannot be assessed. In any case, if the authors have information on coke deposition in fixed- and fluidized-bed reactors, they should compare and explain the results.

- The information obtained with a model compound is not decisive to compare BP with bio-oil as feeds for the FCC unit. Thus, the comparison made must be rewritten, writing the advantages of BP in a more cautious way, because they have not really been proven.

Consequently, the manuscript requires a major revision attending to these suggestions.

Author Response

First of all, the authors express our appreciation to the Editor and anonymous reviewers for taking the time and effort to assess this manuscript. Your insightful comments and suggestions have contributed significantly to the improvement of this manuscript. The authors carefully revised the manuscript to address the reviewer’s concerns, as indicated below (all modifications are indicated in the file manuscript with correction indication).

  1. Presumably, the process of biomass ketalization is an unknown process for the readers of the journal and should therefore be briefly explained.

We would like to thank the reviewer to point out that we should describe the new modification of biomass of the second generation, which we referred to as ketalization, to obtain a new source which we called BP in more detail since it is not that well known. In addition, we should further explain and justify the use of DX in n-hexane as a model reaction more clearly.

This ketalization process is carried out by submitting a second-generation biomass (such as sugar cane bagasse in our case) in the presence of a large amount of ketones at a temperature range of 100 -180 °C in the presence of a small amount of acid as a catalyst. Unlike catalytic pyrolysis, the product obtained is not a pyrolysis “bio-oil”. The product does not contain an aqueous phase (no water is present), is stable, and is not acidic. Since it can be readily transformed into other useful products by typical refinery processes, and to distinguish it clearly from bio-oil of pyrolysis, we named it as “Bio-petroleum”. Detailed examples are given in the references.

Even though the application of ketones on second-generation biomass is indeed an inventive process of our group, the ketalization is a well-known method in organic chemistry in which ketones are used to react and protect active OH groups of a reagent before it is subjected to further transformation. A worker in this field could readily derive modification of this method to suit his specific biomass source and product requirements. At any rate, to better informed the reader, we have added a sentence at the end of the introduction paragraph 1 to describe the ketalization process.

“In short, the ketalization process is carried out by submitting a second-generation biomass (such as sugar cane bagasse in our case) in the presence of a large amount of ketones at a temperature range of 100 -180 °C and a small amount of acid as catalyst.“

  1. In addition, the composition of the bio-petroleum (strange notation) should be explained in greater detail and the selection of DX as a model justified.

The general composition of BP is given in Table 1 of the text. Naturally, the detailed composition of a product of ketalization is a function of the ketalization conditions and the biomass source. In Table S 1 we will give a more detailed example of the components present in a BP produced in a typical condition. Further examples could be found in the references cited.

Once a typical BP contains 80% total ketals and more than half of these ketals are Di-ketals, DX suggests itself as a representative compound to be used for tests in catalytic transformation. The justifications were given at the end of the third paragraph of the introduction:

“Noteworthy to point out that the advantage of using DX as model test could simplify the product analysis, give clearer insight to the reaction steps in the ketal transformation and hence suggest more definite correlation between catalyst properties and the product yields. Using either DX or BP as reactants, very similar catalytic results were observed in a variety of tests. These results are subjects of future publications [1, 2].”

  1. The purpose of using n-hexane together with DX should also be explained.

Indeed, there are a lot of choices for a co-feed that could facilitate or inhibit the transformation of the bio-mass reagent. In the present case, we would like to focus on the transformation of the model molecule DX. Hence, we select a molecule that could mix well with DX, does not hinder the access of DX to the active sites and does not give much cracking product itself.

For a clearer description of the choice of n-hexane we have substituted in the third paragraph of the introduction the sentence “As low reactivity co-reagent, n-hexane was also used” with a more comprehensive one:

“n-hexane was chosen as co-reagent, since it mixes well with DX; being a short saturated hydrocarbon, it provides much less competition for DX to access the active sites. Further, it is much less reactive and gives less products itself in catalytic cracking reaction.”

  1. Likewise, the objective of including the so-called “additive” should be explained, because it seems to be the catalyst embedded in a mesoporous matrix, which is the configuration of FCC catalysts.

Thanks to pointing out the better description of our catalyst, which we had adopted due to its industrial usage in FCC processes. But for this article, we have changed the name of “additive” to “catalyst” and named it as catalysts AD (fresh) and DAD (steamed).

  1. The purpose of steaming the FCC catalyst is to condition its porous structure and acidity, equilibrating the catalyst so that it reproduces its performance (with high gasoline yield) in reaction-regeneration cycles. Consequently, the treated catalyst is called "equilibrated" or "stabilized". “Deactivated” is the universal denomination of the spent catalysts after their use.

Thanks to pointing out the better description of our catalysts pre-treating conditions, we have used this originally to distinguish them from the fresh catalyst since they lost most active sites. We have changed the description simply to “steam treatment” catalyst for a more accurate description.

  1. The performance of catalysts with different zeolite has been studied in cracking bio-oil (Ibarra et al., J. Ind. Eng. Chem., 78 (2019) 372-382). This is a good reference on the effect of shape selectivity and acidity on product distribution.

Many thanks for bringing to our attention of this work of Dr. Alvaro et. al. In fact we have followed some works of Prof. J. Bilbon and Prof. U. Sedran. We will cite this reference in our text when we refer to study of cracking of bio-oil.

            However, again we would point out there very important differences between pyrolysis bio-oil and BP, especially:

  • BP does not contain an aqueous phase, while bio-oil contain 18 -35% water.
  • BP only contains an insignificant amount of acidic component while acetic acid is a major component of bio-oil.
  • The phenolic component of the bio-oil, resistant to cracking is also noted while only small amount of phenolic derivatives were observed in the product of BP cracking under typical conditions.

            From their product analysis, we observed that the deoxygenation of bio-oil in the simulated FCC conditions, were very poor. The amount of oxygenates are much higher than the hydrocarbon forms. Acid and phenols are left as the major components in the gasoline fraction.

            Hence, we will only apply the knowledge derived from the vast literature on the effect of zeolite shape selectivity and acidity on cracking of hydrocarbon and adapted them as explanations of the catalytic results of the present study.

  1. If the authors have information on the acidity of the catalysts, they should include it, because it is essential for understanding the results.

It has not yet been possible to perform acidity analysis with the beta catalyst. However, in Table 2 the acidity of beta zeolite is described.

8.The characteristics of the fluidized reactor should be better explained. And the reasons for the different results in fixed- and fluidized-bed reactors should also be explained.

The fluidized bed reactor is a typical reactor constructed for micro-activity tests (MAT). It as has been fully described in our previous publications [3–5]. The general differences between a fixed bed reactor and a MAT unit has been discussed also in our previous publications using similar feeds [4, 6]. Indeed, aside from the huge difference in catalyst to feed ratio, the important additional difference is the much longer contact time of the fixed bed reactor, acting to the advantage for the conversion of the feed and its intermediate products. This advantage was much attenuated by the rapid deactivation of the catalyst. In spite of this advantage, the large difference in catalyst to feed ratio favored the conversion in the fluidized bed. A general discussion on the differences in the reactors and further ongoing studies is given in section 3.5 of the text.

“Firstly, we would further comment on the differences of the two testing processes. Both processes, FB and FCC demonstrated successful DX conversion into green hydrocarbons. They also showed the benefits of a thermal deactivation treatment to a Beta catalyst, used pure or as the main active component in a catalyst. Both reaction modes can be used complementarily to develop and design catalysts to improve DX and further, BP conversion into target products. FB is less time-consuming compared to FCC. Thus, FB can be used as an initial screening for catalyst properties. FCC usually reduces external diffusion and pressure build-up, but it is more time-consuming than FB [8]. In the cases compared, better deoxygenation, without much coke penalty, was obtained in FCC. Despite the high deoxygenation, these compounds are still present in the conditions used (30 wt.% DX and catalyst/feed =3). However, the catalyst/feed ratio can be adjusted to values as high as 7-8 in FCC. In a separate study, we further varied certain reaction conditions in the FCC process, such as the catalyst/DX ratio, the injection time, and also % DX in the feed mixture [13]. This later study will be reported in another publication, focusing on the process parameters, not on the catalyst properties.

On the other hand, it is expected that under certain conditions, FB can be used to convert BP. For example, when one uses catalysts with less activity, the FB unit may operate with a higher catalyst/load ratio if one could solve the problems with diffusion and pressure build-up. We are testing this possibility by using catalysts with different physical formats, such as pellets or extrudates. In short, we aim at the use of BP to produce drop-in fractions of green- hydrocarbons in both conversion modes, FB and FCC.”

Due to the complex network of the transformation, and further typical differences between the two types of reactors, we think that a detailed explanation is not yet warranted for the results obtained. Further studies in both units are in progress. The study in the FCC unit will be published shortly.

  1. The role of water is very important to attenuate the deactivation of the catalyst in bio-oil cracking (Ibarra et al., Energy Fuels 33 (2019) 7458-7465). Consequently, a high deposition of coke can be expected. Do the authors have information on this regard?

Again, we would point out a very important difference between pyrolysis bio-oil and BP: BP does not contain an aqueous phase, while bio-oil contains 18 -35% water.

Even though water is an important product of de-oxygenation, we observed that it is formed and could be collected at the very beginning of the transformation process, so its effect on on-site competition, deactivation, and coke deposition is not as important as the case of bio-oil cracking. In general, we observed both in the case of increasing the % DX in the feed mixture or when the degree of transformation advanced, both the water and coke increased as products.

  1. The experiments have been carried out with catalyst excess. Consequently, the conversion is almost 100% and the deactivation cannot be assessed. In any case, if the authors have information on coke deposition in fixed- and fluidized-bed reactors, they should compare and explain the results.

The objective is not only transformed this biomass derivative DX, but to obtain bio-hydrocarbons with minimal oxygenates. Hence, only 2 experiments reported here were approaching this stage (with a fresh catalyst in the fixed bed or with a steamed catalyst in the fluidized bed). The results could be further improved. Hence, we would not consider the catalysts used in excess.

As mentioned in the text, we further varied certain reaction conditions in the FCC process, such as the catalyst/DX ratio, the injection time, and also % DX in the feed mixture, focusing on the process parameters, and will be reported in another publication. In this case, we could only obtain indirect information on deactivation from interpretation of the coke yields. (There results are under preparation and will be send to a special issue of Chemical Engineering Research and Design in May!)

By contrast, due to no high coke formation and yet already with high deoxygenation from the catalysts, further studies are being made using the fixed bed reactor. Here, indeed, the formation and catalysts activity and selectivity will be reported as a function of time on stream. The results will be published in due course.

  1. The information obtained with a model compound is not decisive to compare BP with bio-oil as feeds for the FCC unit. Thus, the comparison made must be rewritten, writing the advantages of BP in a more cautious way, because they have not really been proven.

We appreciate this observation. Indeed, our results using mixtures of BP and VGO are under preparation. We anticipate DX is indeed a representative compound of BP. But as we did not submit the results, we modify section 3.5 accordingly as presented below.

The main objective of the present work is to show some variation of the catalysts parameters can affect a promising conversion of the model compound (DX) to bio-hydrocarbon.

This is not a comparison between the BP cracking process and the cracking of bio-oil. Since, the generation of biofuel from biomass has to be considered in a global sense, from the biomass. BP is produced from a mild process with no carbon loss, and pyrolysis bio-oil already has about 20% of carbon loss. Bio-oil generated has an acidic and unstable property.

Bio-oil to be deoxygenated is difficult. If we are obliged to make a comparison, we would point out that the ease of de-oxygenation of BP is a clear advantage. (in this work using cat/oil 3 in a MAT unit ). The co-processing of BP with VGO in mixtures from 0 to 75% of BP is being submitted for publication.

Now let us discuss the overall picture linking our previous results of renewable production from biomass, extending the results of DX to bio-petroleum, BP. The biomass transformation chain into hydrocarbons occurs in two steps: 1 - from biomass to BP with no carbon loss, and 2 - from BP, a stable non-acidic chemical platform, to hydrocarbons.

Initially, DX was used as a feed or a representative model compound of BP, as it is present in BP up to 30 wt.% [7]. Our further experimental results comparing the conversion of DX and BP under comparable conditions showed indeed the BP transformation closely parallels that of DX: not only in the cracking process but further in hydro-deoxygenation processes[3, 6, 8]. From one of these studies, a work of co-processing BP with VGO over equilibrium catalysts, with a mixture up to 75wt.% BP is being submitted for publication.

Though the biomass conversion to fuel process should not be compared only in the conversion of derivative, however, it is worth mentioning that conversions of bio-oil in typical FCC conditions generally leave a high amount of acidic and phenolic oxygenates. Further, co-processing bio-oil with mineral oil is usually converted in concentration up to 20 wt.%.[9–11].

References

  1. Souza MO de (2022) Produção de hidrocarbonetos renováveis através da hidroconversão de cetais de D-xilose utilizando catalisadores de Paládio em zeólita Beta. Universidade Federal do Rio de Janeiro
  2. Cardoso C de S (2022) Conversão do Acetal de D-Xilose em Bio-Hidrocarbonetos por Craqueamento Catalítico com Zeólita Beta. Universidade Federal do Rio de Janeiro
  3. Pinto JFR, Lam YL, Pereira MM, et al (2019) Green-aromatic production in typical conditions of fluidized catalytic cracking. Fuel 254:115684. https://doi.org/10.1016/j.fuel.2019.115684
  4. Pinto J, Pedrosa I, Linhares C, et al (2019) Ketal Sugar Conversion Into Green Hydrocarbons by Faujasite Zeolite in a Typical Catalytic Cracking Process. Front Chem 7:1–14. https://doi.org/10.3389/fchem.2019.00720
  5. Pereira SC, Souza M, Pinto J, et al (2020) Sugar ketals as a platform molecule to overcome the limitation of converting biomass into green-hydrocarbons in a typical refinery. Sustain Energy Fuels 4:1312–1319. https://doi.org/10.1039/c9se00379g
  6. Cardoso C, Lam YL, San Gil RAS, et al (2022) Conversion of sugar diacetyls to bio-hydrocarbons by the catalytic cracking in a fixed bed with fresh and deactivated Beta zeolite. Catal Commun 171:106519. https://doi.org/10.1016/j.catcom.2022.106519
  7. Dos Santos DN, Pedrosa I V., Fernandes CRR, et al (2020) Catalytic sugarcane bagasse transformation into a suitable biocrude for hydrocarbon production in typical refinery processes. Sustain Energy Fuels 4:4158–4169. https://doi.org/10.1039/d0se00220h
  8. Souza MO, Pereira SC, Lau LY, et al (2021) Hydrodeoxygenation of Xylose Isopropylidene Ketal Over Pd/HBEA Catalyst for the Production of Green Fuels. Front Chem 9:1–12. https://doi.org/10.3389/fchem.2021.729787
  9. Varma AK, Mondal P (2017) Pyrolysis of sugarcane bagasse in semi batch reactor: Effects of process parameters on product yields and characterization of products. Ind Crops Prod 95:704–717. https://doi.org/10.1016/j.indcrop.2016.11.039
  10. Gayubo AG, Aguayo AT, Atutxa A, et al (2005) Undesired components in the transformation of biomass pyrolysis oil into hydrocarbons on an HZSM-5 zeolite catalyst. J Chem Technol Biotechnol 80:1244–1251. https://doi.org/10.1002/jctb.1316
  11. Pinho A de R, de Almeida MBB, Mendes FL, et al (2017) Fast pyrolysis oil from pinewood chips co-processing with vacuum gas oil in an FCC unit for second generation fuel production. Fuel 188:462–473. https://doi.org/10.1016/j.fuel.2016.10.032

Round 2

Reviewer 3 Report

The authors have carried out the modifications suggested in the first version and the revised version fulfils the requirements for publication.